# An Evaluation of the Physical and Chemical Stability of Dry Bottom Ash as a Concrete Light Weight Aggregate

**DOI:** 10.3390/ma14185291

**Published:** 2021-09-14

**Authors:** Jinman Kim, Haseog Kim, Sangchul Shin

**Affiliations:** 1Department of Architectural Engineering, Kongju National University, 275 Cheonan-daero, Cheonan City 330-717, Chungcheongnam-do, Korea; jmkim@kongju.ac.kr; 2Environment-Friendly Concrete Research Institute, Kongju National University, 275 Cheonan-daero, Cheonan City 330-717, Chungcheongnam-do, Korea; bravo3po@kongju.ac.kr

**Keywords:** bottom ash, coal ash, dry process, light weight aggregate

## Abstract

Compared to the bottom ash obtained by a water-cooling system (wBA), dry process bottom ash (dBA) makes hardly any unburnt carbon because of its stay time at the bottom of the boiler and contains less chloride because there is no contact with seawater. Accordingly, to identify the chemical stability of dBA as a lightweight aggregate for construction purposes, the chemical properties of dBA were evaluated through the following process of the reviewing engineering properties of a lightweight aggregate (LWA). Typically, river gravel and crushed gravel have been used as coarse aggregates due to their physical and chemical stability. The coal ash and LWA, however, have a variety of chemical compositions, and they have specific chemical properties including SO_3_, unburnt coal and heavy metal content. As the minimum requirement to use the coal ash and lightweight aggregate with various chemical properties for concrete aggregate, the loss on ignition, the SO_3_ content and the amount of chloride should be examined, and it is also necessary to examine heavy metal leaching even though it is not included in the standard specifications in Korea. Based on the results, it is believed that there are no significant physical and chemical problems using dBA as a lightweight aggregate for concrete.

## 1. Introduction

Coal ash is produced by drying and pulverizing bituminous coal in a pulverizer at a power plant and burning the pulverized coal in a boiler. About 15 to 45% of the pulverized coal is mainly collected from a dust collection system or the bottom of the boiler. Bottom ash refers to the ash that has fallen into the clinker hopper at the bottom of the boiler. After it falls to the bottom of the boiler and becomes accumulated in the hopper, it is ground by a grinder. The amount of the bottom ash produced accounts for around 10 to 25% of the total coal ash [1,2].

Approximately 40% of the bottom ash produced from Korea is recycled, while the remaining 60% is landfilled. Most of the bottom ash is recycled into landfill aggregates, which have low added value. Bottom ash has a high chloride concentration, a high unburned carbon content, and a high moisture content due to the use of seawater in the cooling process, and the chloride contained in the bottom ash causes corrosion of the embedded reinforcing bars. Moreover, the non-constant condition causes technical difficulties when using bottom ash and acts as an impediment to adding value as it cannot be used in a dry state [3,4].

The unburned carbon in bottom ash is partially used as a low value-added material such as a filling material or a subsidiary material for cement, whereas its uses in high value-added products such as ready-mixed concrete (unburned carbon content of less than 5%) are only marginal. As for domestic technologies related to bottom ash, there have mostly been studies on its uses as a raw material for cement, binder, or construction material, the pretreatment of bottom ash, and its catalytic potential and other applications. Due to the characteristics of bottom ash, there are limits to its use as a cement, binder, an aggregate or a lightweight aggregate (LWA) material. Accordingly, it is mainly used as an aggregate (low value-added use) for filling and covering sports fields and drainage areas. In order to promote the recycling of bottom ash, research has been carried out to determine its dissolution potential based on the content of harmful substances and dissolution characteristics so that it could be used as a fill and cover material, but a negative perception of bottom ash is prevalent due to the uncertainty of the potential environmental impact. For the purpose of resolving the issue of unstable quality, domestic power plants have switched from the wet process to the dry process when it comes to bottom ash treatment so as to lower the content of moisture, salt and unburned coal, and are striving to boost the recycling rate and expand the scope of recycling. Dry bottom ash (dBA) is an industrial byproduct emitted from thermal power plants, and a detailed description is provided in Section 2.1.

As of 2018, the amount of bottom ash is 1.35 million tons in Korea, of which dBA is estimated to be 30%, which is about 400,000 tons. The outer section of dry bottom ash is characterized by an open-cell porous structure, so it is lightweight with a high absorption rate, high hardness, and excellent physical properties, making it suitable as a permeable aggregate [5,6].

In this study, the physical and chemical properties of dry bottom ash produced in Korea using the dry method were analyzed, and the possibility of using dry bottom ash as LWA for lightweight concrete was experimentally examined.

## 2. Preceding Research on the dBA

### 2.1. The Discharging Process of dBA

Dry bottom ash (hereinafter referred to as dBA) is an industrial by-product discharged from thermal power plants in Korea, as shown in Figure 1. dBA discharged from a dry process is moved without coming into contact with water by the primary clinker conveyor. After passing through the primary conveyor, it is pulverized to a certain size and then finally cooled on the secondary conveyor and discharged after the secondary crushing process. The passage time of the primary and secondary conveyors is 40 to 60 min each, and most of the unburned coal is burned during residence on the conveyor. The particle size of the discharged dBA varies greatly in range, from more than 100 mm to less than 1 mm. It is quite irregular in shape, as shown in Figure 2 [7].

### 2.2. Pore Properties of dBA

Figure 3 is a photograph showing the surface and internal pore properties of dBA observed with a scanning electron microscope (SEM). It shows that the dBA surface is composed of pores on a thin film and that the outside and the inside are connected due to partial destruction. On the inside, there are pores of various sizes that exist independently or continuously. Figure 4 is a graph showing the porosity according to the particle size. While the total porosity varies depending on the size of dBA, it has been found to be 50 to 60%, and there was a positive correlation between the size of the aggregate and the number of closed pores.

### 2.3. Physical Properties of dBA

Table 1 shows the density, absorption rate, unit volume mass, and the percentage of absolute volume as an examination of the physical properties by dBA size. The density and absorption rate of dBA were measured using the standard test method for the density and absorption of a coarse aggregate (KS F 2503) and the standard test method for the density and absorption of a fine aggregate (KS F 2504). The unit volume mass and percentage of the absolute volume of dBA were measured by using the standard test method for the bulk density and the percentage of absolute volume in an aggregate (KS F 2505) [8,9,10].

An analysis of the correlation between density and absorption rate shows that a decrease in particle size is accompanied by an increase in density and absorption rate, as shown in Figure 5. This is opposite to the inverse relationship between density and absorption rate in general. Figure 6 shows the correlation between the density and absorption rate of general aggregates. The difference between dBA and general aggregates is deemed to be due to the large number of pores present in the matrix in the former. In the case of density, the larger the particle, the lower the number of pores and the higher the absorption rate, and this is judged to be because there are a relatively larger number of continuous pores when there are smaller dBA particles, resulting in an increased amount of moisture penetrating the dBA. Moreover, while the unit mass increases, the percentage of the absolute volume tends to decrease. It is thought that this is because the number of open pores increases as the particle size decreases, as shown in Figure 4. Furthermore, it has been indicated that the rate of sphericity or flatness of the aggregate does not improve even when the aggregates become smaller [11,12].

### 2.4. Feasibility of dBA as a Construction Material in Relation to its Physical Properties

In a preceding study, the following conclusions were obtained as a result of examining the physical properties of dBA, a by-product in the electric power industry, as a LWA (Light Weight Aggregate) for construction [13].

As for the shape of dBA, it is structurally weak due to the sharp and angular edges and the flat and elongated shape, but unlike the surface, where there are open cells, the inside features closed cells and is relatively high in hardness.

In addition, although the particle density decreases along with a decrease in the particle size, the apparent density tends to increase slightly. This is because large cells are destroyed as the particles become smaller. Since closed cells that are larger than 100 μm are destroyed and turn into open cells as a result of the reduced particle size, there is an increase in the open porosity, a decrease in the closed porosity and total porosity, and an increase in the absorption rate.

Therefore, based on the findings of the previous study, it appears that dBA can be used as a LWA and has excellent properties in terms of weight and thermal insulation if the relatively weak surface is removed and it is processed into a near-spherical shape [14].

Table 2 shows the advantages and drawbacks of various aggregates including dBA used in this study. The wBA discharged from the wet process is difficult to recycle due to problems such as high unburned carbon, high chloride, and moisture contents. Natural aggregates are relatively high-quality aggregates that have been used for a long time, but they are related to environmental problems and resource depletion. In the case of the artificial lightweight aggregate, it is a product with a constant quality with a spherical shape and a uniform particle size because it is produced in a factory; however, it has the disadvantage of high manufacturing costs and greenhouse gas emissions due to the calcination process. In addition, in the case of Korea, the cost is high because most of it depends on imports. On the other hand, dBA has the potential to be applied as an alternative material for lightweight aggregates, owing to the low content of unburned carbon, chloride and SO_3_. Nevertheless, the structural weakness caused by the irregular shape and high absorption is a problem to be overcome.

## 3. Experimental Plan and Method

### 3.1. Experimental Plan

Table 3 shows the experimental plan of this research. This study was conducted with the aim of determining the possibility of using dBA, which has been confirmed to have appropriate physical properties as LWA, for concrete manufacturing by evaluating its chemical properties.

The chemical properties of dBA were analyzed based on X-ray fluorescence (XRF) to check the oxide composition and SO_3_ content, X-ray diffractometry (XRD) to examine the mineralogical properties, loss on ignition to measure the amount of unburned coal as well as chloride content and the pH, and the heavy metal leaching test.

The dBA evaluated in this study was bottom ash discharged from a dry process at the B Thermal Power Plant operated by J Power in Korea. For a comparison, fly ash (FA), wet bottom ash (wBA), and four domestic and foreign artificial LWAs were examined as well. The experimental plan is shown in Table 2.

### 3.2. Shapes of Various Artificial Aggregates

Figure 7 shows digital camera and SEM images of the samples used in this study.

As described above, dBA has sharp edges and is flat, elongated, and irregular in shape. In the case of wBA, it has the appearance of a crushed aggregate (crushed stone), and the corners are relatively round compared to dBA. On the other hand, FA has a near-perfect spherical shape.

LWA-1 is an artificial LWA produced in Korea that has a generally round shape, while LWA-2 is a product from the United States and has a relatively angular shape compared to LWA-1. In the case of LWA-3, it is a product made in Japan and is an intermediate between LWA-1 and 2 in terms of shape, while LWA-4 is a product made in China and its shape is closest to a sphere.

### 3.3. Test Method

Aggregates used as coarse aggregates for concrete are generally made from river gravel and crushed gravel with stable physical and chemical properties. Coal ash and LWA, on the other hand, have various chemical compositions and have specific chemical properties such as SO_3_, unburned carbon, and heavy metal content. As such, in order to use coal ash and a LWA with various chemical properties as aggregates for concrete production, the loss on ignition, sulfur trioxide content and chloride content among other factors are reviewed as a minimum requirement in Korea. Plus, in order to use dBA as industrial waste generated from power plants, it is necessary to determine its environmental impact. Therefore, this paper presents a review of the leaching of heavy metals from dBA, and the experimental items and evaluation methods were as follows.

#### 3.3.1. Oxide Composition by XRF

This is a method of analyzing the characteristic energy emitted from the samples using a Rigaku ZSX Primus X-ray Fluorescence Spectrometer (XRF-WDX) after grinding the samples to 80 μm or less in size in order to examine their chemical composition. The composition of the constituent elements was quantitatively evaluated.

#### 3.3.2. Mineralogical Analysis by XRD

In order to determine the mineral phase of coal ash and the artificial lightweight aggregate and the bonding state of minerals, the samples pulverized to 8 μm or less in size were examined by X-ray diffraction using Rigaku DMAX2000. This method involves analyzing the diffraction image obtained when the X-ray is incident on the crystal surface at a specific angle and scattered by the atomic layer in the crystal surface which satisfies Bragg’s law.

#### 3.3.3. Chloride Content

The chloride analysis was conducted in accordance with KS F 2713—Testing method for analysis of chloride in concrete and concrete raw materials [15].

#### 3.3.4. Unburned carbon

To test the amount of unburned carbon in the test specimen, an experiment was conducted based on KS L 5405—fly ash [16]. According to the instructions, 1 g of the sample was measured to 0.1 mg in a crucible, ignited for 15 min in an electric furnace at 975 ± 25 °C, and cooled in a desiccator before the mass was measured. Ignition was repeated for 15 min until a constant mass was reached. Moreover, the PerkinElmer Pyris 1 TGA equipment was used to examine the rate of change in weight according to temperature.

#### 3.3.5. Potential of Hydrogen

The pH test was conducted according to KS M 0011—the method for the determination of the pH of aqueous solutions, and a pH meter with a precision of ±0.01 was used [17]. In this test, 200 g of the sample was immersed in 200 g of normal water and distilled water and then the pH of the aqueous solution was measured for 48 h. Moreover, in order to examine the initial pH and leaching characteristics, the samples were stirred for 30 s before measurement.

#### 3.3.6. Heavy Metal Leaching Test

The heavy metal leaching test was conducted by applying two test methods in parallel: the Waste Process Test Method (Ministry of Environment of Korea Notice No. 2011-160 [18]) and the Official Soil Pollution Test Method (Ministry of Environment of Korea Notice No. 2013-113 [19]). Both tests use the ion detection method and the methods of preparing the test solution are identical. To be more specific, the sample and the solvent were mixed in a 1:10 ratio to create at least 500 mL of the mixture, after which the mixture was shaken for about 6 h at a rate of around 200 times per minute in a 4–5 cm shaker. Then it was filtered for the filtrate to be used as the test solution.

#### 3.3.7. Minor Inorganic Compounds by ICP

In order to measure the concentrations of inorganic elements in coal ash and artificial LWA, an inductively coupled plasma optical emission spectrometer, Optima 5000 DV from GE, was used. A total of 0.25 g of the sample was pretreated with 0.5 mL of hydrofluoric acid (HF) and 5 mL of nitric acid (HNO_3_) each and diluted by 200×.

### 3.4. The Physical Properties of Various Artificial Aggregates

Table 4 shows a prior study that measured the physical properties of fly ashes and LWAs. The absolute dry density was 2.1 g/cm^3^ of FA which was the highest among the aggregates in the analysis. LWA 2, 3, and 4 showed similar values of 1.50~1.53 g/cm^3^, and LWA-1 was 1.70 g/cm^3^ which was relatively slightly higher. dBA showed an absolute density of 1.72 g/cm^3^ and among the overall aggregates in analysis it showed moderate values.

As for the absorption, LWA-2 was 2.36%, which was relatively very low compared to the other aggregates with values of around 10%.

From the measurements of the unit volume mass and performance rate, dBA had a relatively low performance while LWA-1 showed the highest performance. This is due to the high density of LWA-1 and the fact that it is spherical with a smooth surface. Therefore, it has a high-performance rate while dBA has a relatively low sphericity, and the pores are exposed on the surface.

## 4. Test Results and Discussion

### 4.1. Oxide Composition

The chemical composition of coal ash varies according to the type, quality, and combustion characteristics of coal, and it is as shown in Figure 8. SiO_2_, which is the primary component, accounts for 50%, while Al_2_O_3_ accounts for 20%, CaO accounts for 2~3%, and MgO accounts for 1~2%. From the perspective of utilizing coal ash, alkali cations, such as Ca, Mg, K, and Na, in addition to Fe and S, are important evaluation factors, and coal ash is classified according to the content ratio of these elements [20]. As for the chemical composition of typical bituminous coal, a type of coal ash generated from domestic power plants, SiO_2_ and Al_2_O_3_ account for at least 85%, CaO accounts for 2~3%, MgO accounts for around 1%, and SO_3_ accounts for around 1~2% [21]. Soluble SiO_2_ in coal ash combines with Ca(OH)_2_ generated during cement hydration at room temperature to form insoluble calcium silicate (CaO·SiO_2_·nH_2_O) that is stable in form, thereby causing a pozzolanic reaction that increases the long-term compressive strength of concrete. Moreover, when fly ash is used as a subsidiary material in cement, it behaves as a fusing agent (Al_2_O_3_, Fe_2_O_3_) that lowers the melting point during the clinker reaction, which is why it is a crucial raw material in the manufacturing process of cement [22,23,24,25].

In this study, an XRF analysis was carried out to determine the chemical composition of each sample, and the results are shown in Table 5. The composition analysis of dBA showed that the largest amount of compound found was SiO_2_, followed by Al_2_O, Fe_2_O_3_, CaO, and MgO at 54.9, 19.9, 13.3, 5.7, and 1.9 wt.%, respectively, with a total content of 96%.

In the case of wBA, FA, and LWA-1 to 4, the composition ratio was found to be similar to that of dBA with SiO_2_, Al_2_O, Fe_2_O_3_, CaO, and MgO accounting for more than 90% of the content, despite slight differences. These compositions were found to be similar to that of clay, which is about 50 to 60% SiO_2_, 15 to 18% Al_2_O, 5 to 7% Fe_2_O_3_, and 1 to 7% CaO.

### 4.2. Mineralogical Analysis

#### 4.2.1. Quartz

The mineral crystal form of coal ash provides important information on the reactivity and hardening necessary for the effective utilization of coal ash. The quartz and alumina components of coal are transferred to mullite (3Al_2_O_3_·2SiO_2_) or cristobalite (SiO_2_) at high temperatures to transition into a crystalline state. When there is a crystalline phase such as Fe_2_O_3_, only 20 to 30% of the coal ash is in a crystalline state and the rest is present in a glassy state. Except for quartz, which does not undergo chemical deformation at high temperatures and instead mostly retains its original shape, most of the minerals in coal are generally broken down during combustion [26,27]. In other words, clay minerals lose water and are dissociated into various types of glassy or amorphous components in the cooling process. Among coal ash, quartz particles that are the size of silt and clay contained in coal remain as they are, irrespective of the melt that forms together with other inorganic matter during combustion, and the quartz content remains in all coal ash without being significantly affected by the type or composition of the coal ash. Mullite, which is directly crystallized from the melt when coal ash particles start to cool or are formed as a result of the crystallization of the glassy material at high temperatures in the furnace, is the main crystalline state of the low-calcium coal ash and is independent of the cement reaction [28].

Figure 9 shows the X-ray diffraction analysis of dBA, wBA, FA, and LWA-1 to 4. It was found that quartz (SiO_2_), not cristobalite (SiO_2_), was the main mineral of both the coal ash and the four kinds of artificial LWA. This is because quartz (SiO_2_) can be generated at low temperatures of 200 °C or below and at high temperatures of 2000 °C or above depending on the pressure, but cristobalite (SiO_2_) is only generated at low pressures of 1 atm and in a temperature range of 1500 to 1700 °C. In the case of boilers used at domestic thermal power plants, the pressure is relatively high, and the temperature is maintained in the range of 1500 ± 200 °C, which is probably why the quartz (SiO_2_) state is maintained. Even in the case of the artificial LWA, it is fired and foamed using a rotary kiln, and it is believed that the quartz state is maintained because the firing temperature is around 1100~1300 °C during this process [29,30].

#### 4.2.2. Sulfate

In the case of bottom ash, there is a difference in the content of sulfur trioxide (SO_3_) depending on the coal type and combustion efficiency. Generally, oxides of sulfur are considered to be stable because they are dispersed in the glass aggregate that has been rapidly cooled, as in the case of bottom ash, but if sulfur oxides are present in large amounts, they react with C_3_A (3CaO·Al_2_O_3_) among the minerals constituting cement and generate ettringite, the expandable hydration product. This in turn causes sulfate erosion, which can be a problem in concrete structures [31].

According to KS F 2534:2009—Lightweight aggregate for structural concrete, there are restrictions on the sulfur trioxide content in the case of artificial and natural LWAs, but in the case of bottom ash LWA, it must be no more than 0.8%. This is because some of the coal ash discharged from bituminous coal-fired power plants contains sulfur trioxide. As shown in Figure 10, the sulfur oxide content was found to be 0.06 wt.% in the case of dBA and 2.9 wt.% in FA, while wBA recorded the highest sulfur oxide content at 4.29 wt.%. As such, it can be seen that dBA conforms to KS F 2534, whereas wBA does not. In addition, the fired artificial LWA was found to have a relatively stable sulfur oxide content of 0.1 to 1.0 wt.%.

#### 4.2.3. Chloride

The sulfur trioxide (SO_3_) content was analyzed using the XRF result values of each sample. In Korea, dBAs are generally combusted in a boiler and settled in a tank that is filled with seawater at the bottom, after which they are crushed to a certain size and then transferred to an ash treatment plant using a water pipe to be buried. This is why dBAs have a substantially high salt content.

Due to the high salt content, dBAs can cause corrosion of rebars and can seriously reduce the durability of the reinforced concrete structure. For this reason, there is a limit to the amount of chloride that can be contained in the raw materials used in structural concrete.

KS F 2526—Aggregates for concrete and KS F 4570—Bottom ash aggregate for the precast concrete product, for example, require that the salt content be no more than 0.04% and 0.025%, respectively [32,33]. In addition, KS F 2534—Lightweight aggregate for structural concrete, limits the salt content of fine aggregates made from bottom ash to no more than 0.025% to ensure quality.

As a result of measuring the chloride content of the various test materials, it was found that dBA generated by the dry process does not contain any chloride, as shown in Figure 11. However, in the case of wBA generated by the wet process, the chloride content was measured to be 0.038%, exceeding the limit specified in the KS. This is believed to be because wBA is cooled and transported using seawater, as described above.

Moreover, Cl ions (approx. 0.01%) were detected even in the artificial LWA, which suggests the need to pay attention to the chloride content even in the case of fired artificial LWA.

#### 4.2.4. Unburned Carbon

Bottom ash produced by the wet process contains a large amount of unburned carbon because it is cooled immediately when it falls to the bottom of the boiler. If unburned carbon is evaluated based on loss on ignition, the loss on ignition of the bottom ash from the wet process in Korea varies depending on the coal type and combustion efficiency, but it is generally known to be about 1.5 to 15%. If the loss on ignition is high, the unburned carbon adsorbs the entraining air introduced to improve the durability of the concrete, and this adversely affects the durability of the concrete. Therefore, the unburned carbon content is limited to 5% for concrete materials used in powder form. In the case of aggregate-like materials, there is no limit when it comes to the unburned carbon content, but in case of using bottom ash aggregates with high loss on ignition, more admixtures are required to ensure the same workability and air volume, making concrete production less economical. Unburned carbon powder with large particles may cause concrete delamination and thus may serve as a factor deteriorating the quality of the concrete surface, causing problems in ensuring quality [34,35].

A comparison of the unburned carbon content of dBA and FA is shown in Figure 12. The samples used were FA and dBA discharged from Facilities 7 and 8 of the B Thermal Power Plant operated by J Power in Boryeong, Chungcheongnam-do Province. It was found that the unburned carbon content of FA exceeded the requirement (5%) specified in KS L 5405—Fly ash. On the other hand, dBA was found to have an unburned carbon content of no more than 2% during the 6-month period and was found to exhibit stable qualities satisfying the requirement (5% or less) specified in KS F 2534:2007—Lightweight aggregate for structural concrete.

Figure 13 shows the results of a gravimetric analysis of dBA coal ash and LWAs using DT-TGA. dBA was found to have a low unburned carbon content with a weight reduction rate of less than 1%, as was the case during the 6-month tracking period. Likewise, in the case of artificial LWAs, although LWA-4 recorded a relatively high value, all of the samples underwent a small degree of weight reduction of less than 1%. This is due to the fact that artificial LWA is put through a sintering process for the foaming of the aggregates at around 1100 to 1300 °C, and most of the unburned carbon is burned in this temperature range [36,37].

#### 4.2.5. Hydrogen Exponent (pH)

Aggregates used in the construction industry are mostly stored outdoors, as they are used in large amounts and are large in volume. Aggregates stored outside in this way lead to leachates due to rainwater, etc., which makes it necessary to consider the impact on the surrounding environment. Unlike natural aggregates, crushed stone aggregates are usually chemically stable and do not cause environmental problems. It has been pointed out that recycled aggregates and slag aggregates may result in strong alkaline leachates and may cause environmental problems [38,39].

Therefore, the possibility of environmental problems caused by leachate was examined by measuring the pH of domestic and foreign artificial LWAs and bottom ash. For the experimental method, KS F 2013—Method of measuring the pH of soil was referenced, and the prescribed test method was modified to suit the purpose of this study [40]. Two types of water were used in the measurement: tap water and distilled water. The artificial LWA and dBA samples were immersed in the two types of water in a weight ratio of 1:1, and the changes in pH over 48 h were measured.

Figure 14 and Figure 15 show the results of the measurement of the pH using tap water and distilled water, respectively. Figure 14 shows that the initial pH of the dBA aggregate sample was similar to that of the artificial LWA, but the pH increased after 3 h of immersion before it stabilized at 8.61 after 48 h. Generally, ash represents basicity. WBA has a high unburnt carbon content and dBA has a low unburnt carbon content. Therefore, dBA is relatively highly basic and shows a somewhat higher pH value. As such, dBA has a low level of alkalinity, but this is not expected to adversely affect the surrounding environment. However, when dBA was pulverized into powder, the pH was found to be the highest at 9.4 at the beginning of immersion, and although it decreased somewhat to 9.15 after 48 h, it was still the highest among all the samples. This is believed to be due to the fact that the specific surface area was increased by pulverization. Generally, when the particle size is smaller and the specific surface area is high, ion elution becomes more active, so the pH value can vary according to the ionization rate according to the specific surface area of the same material. Therefore, care should be taken not to store dBA outdoors when it is in powder form.

In the case of the four types of artificial LWA under comparison, their pH levels were found to be stable at the beginning of the measurement and after 48 h. The pH was 7.4 to 7.5 initially and increased to about 7.8 to 8.6 after 48 h.

The results of the experiment in which the samples were immersed in distilled water for pH measurement are shown in Figure 15. Since the impurities that could interfere with the leaching of ions were removed, the initial pH measurements were all higher than those of the samples immersed in service water. However, the results were similar to their counterparts after 48 h due to ion stabilization.

Moreover, the reason that the pH increased after immersion for all the test specimens was deemed to be due to the leaching of alkali ions, Na^+^, Mg^++^, K^+^, Ca^++^, etc.

#### 4.2.6. Heavy Metal Leaching Test

In order to use bottom ash as a structural aggregate, it is necessary to measure the environmental pollutants that can possibly be leached prior to understand their impact on the environment before applying them in the field. In each country, there are laws regulating the amount of pollutant that can be leached during recycling and specifying whether bottom ash can be recycled. Bottom ash is composed of SiO_2_, Al_2_O_3_, CaO, Fe_2_O_3_, etc., but it also contains trace amounts of heavy metals [41,42]. Column leaching tests have shown that some of the heavy metals tested according to the drinking water quality standards of Korea, the USA, the EPA, and the WHO were below the baseline or the detection limit. On the other hand, the Pb and Zn levels exceeded the baseline, but the possibility of causing water pollution is low, as they have recorded no more than 1 PVE in all dissolution experiments. Clause 2.3 of KS F 2534:2009—Lightweight aggregate for structural concrete prescribes the content requirements for Pb, Cd, Cr6^+^, Cu, Hg, As, etc. Accordingly, in this study, a heavy metal leaching test was conducted on the bottom ash discharged from the dry process. In the case of artificial LWAs that are distributed in the market, the heavy metal leaching test was not performed.

The heavy metal leaching test was carried out using two methods specified in the Official Waste Test Standards (Ministry of Environment of Korea Notice No. 2011-160, Type–I) and the Official Soil Pollution Test Standards (Ministry of Environment of Korea Notice No. 2013-113, Type– II) and test results are shown in Table 6 and Table 7. When the official waste test method was applied, dBA was found to be stable, as Pb, Cd, Cr6^+^, Cu, Hg and As were not detected. Also, when the official soil pollution test method was applied, there was partial leaching of five heavy metals Pb, Cd, Cu, Hg, and As, but not Cr6^+^, but leaching occurred in very small amounts compared to the level that would cause concerns of soil pollution.

#### 4.2.7. Minor Inorganic Compounds by ICP

Heavy metals and pollutants may be present on the surface of the sample in a chemically or physically adsorbed state or sometimes in a form in which the pollutant itself is mixed in the sample. Ultimately, in order to protect the soil environment and prevent pollution, there needs to be information on the chemical reactions that can occur between the sample and heavy metals and the characteristics of adsorption to the surface of the sample [43].

Table 8 shows the results of an ICP analysis of coal ash and an artificial LWA. Among the six heavy metals, Pb, Cr, and Cu were found at 0.1, 1.0, and 515 mg/L, respectively, in dBA, whereas Cd, Hg, and As were not leached [44]. Moreover, alkali metals and alkaline earth metals, Na, K, Mg, and Ca, were found at concentrations of 1.2, 34,770, 10,775, and 22.4 mg/L, respectively, and the leaching of these alkali ions was considered to be the cause of the increased pH levels examined above.

Even in the heavy metal content analysis, Cd, Hg, and As were not detected, while the Pb and Cr content was similar to that of an artificial LWA. In the case of the Cu content, it was found to be relatively high, but the amount leached in the leaching test was below the baseline. Thus, it was deemed that there will not be any major problems in using dBA as aggregate for concrete manufacture.

## 5. Conclusions

In this study, the chemical properties of bottom ash produced by the dry process, which is a by-product of the electric power industry, were examined following an analysis of its physical properties to check the possibility of its use as a lightweight aggregate, and the following conclusions were reached as a result:(1)An oxide analysis of dBA showed a composition similar to that of wBA, FA, and four types of LWA, and a mineral analysis showed that the primary component was quartz (SiO_2_), and in the case of dBA, wBA, and FA, the presence of Fe_3_O_4_ was clearer.(2)As for the SO_3_ content, it was found in significantly lower amounts in dBA compared to wBA, FA, and LWA-1 to 4, and the chloride content was also lower in dBA compared to the existing artificial LWAs. Thus, it is believed that there will not be any concerns of expansion caused by SO_3_ or corrosion of rebar by chloride.(3)As a result of measuring the unburned carbon content, it was found that dBA would be a stable aggregate for concrete production, as it has a relatively very low unburned carbon content compared to wBA, FA, and artificial LWAs.(4)In a leaching test carried out in accordance with the Official Waste Test Standards, the six major heavy metals were not detected. In contrast, Pb, Cd, Du, Hg, and As were detected when the test was conducted according to the Official Soil Pollution Test Standards, but they were all below the baseline.(5)An ICP analysis showed that some of the alkali metals and alkaline earth metals were found in larger amounts in dBA than in artificial LWAs, wBA, and FA, thereby resulting in higher pH levels, and Pb, Cr, and Cu were detected in a heavy metal leaching test. However, the heavy metals detected were found at levels below the baseline, based on which it was judged that there will not be any problems in real-life application.(6)Based on the above results, it is believed that there are no significant physical and chemical problems in using dBA as a lightweight aggregate.(7)This paper physically and chemically examined the possibility of using dBA as aggregate for concrete and it is intended that its characteristics by using it in mortar and concrete in the future will be examined.

## Figures and Tables

**Figure 1 materials-14-05291-f001:**
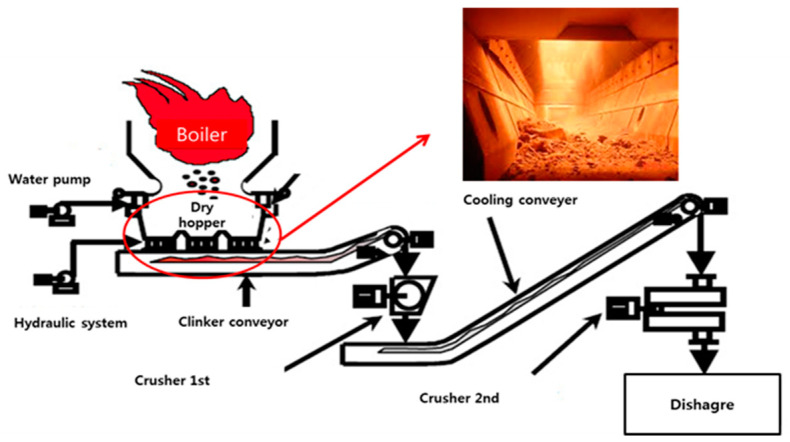
Bottom ash discharging system by the air-cooling process.

**Figure 2 materials-14-05291-f002:**
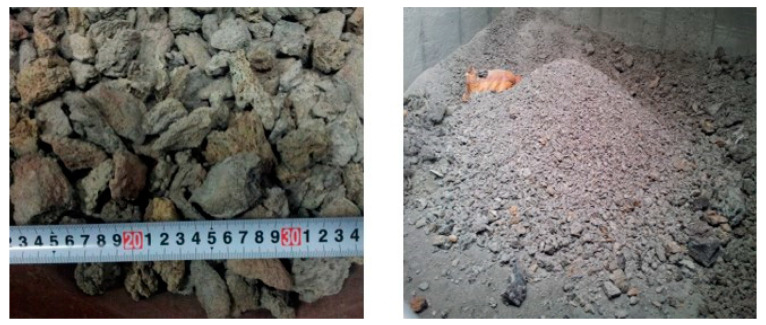
Shape of dBA [7].

**Figure 3 materials-14-05291-f003:**
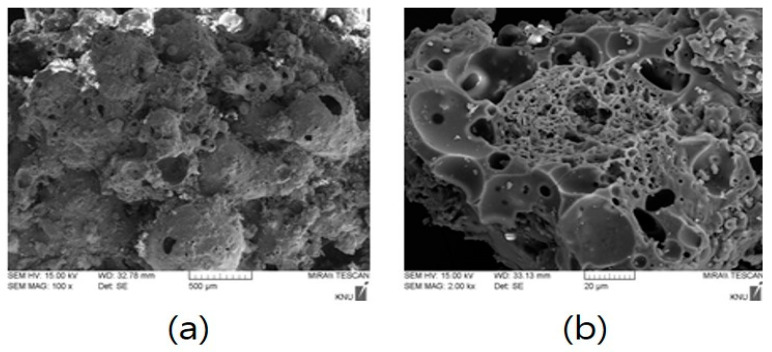
Pore properties of dBA; (**a**) Surface porosity of dBA; (**b**) Internal porosity of dBA.

**Figure 4 materials-14-05291-f004:**
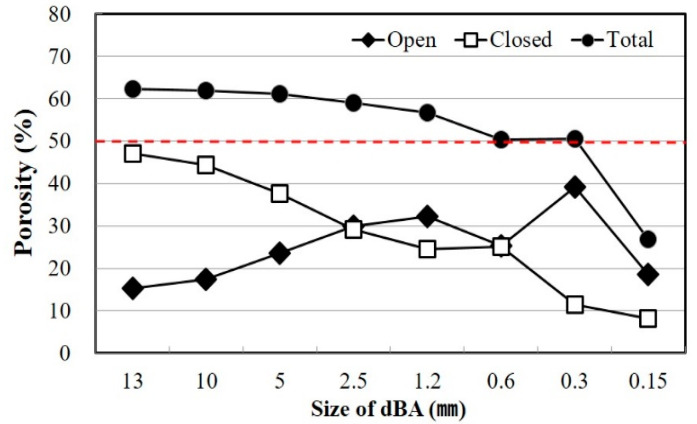
Pore distribution of dBA.

**Figure 5 materials-14-05291-f005:**
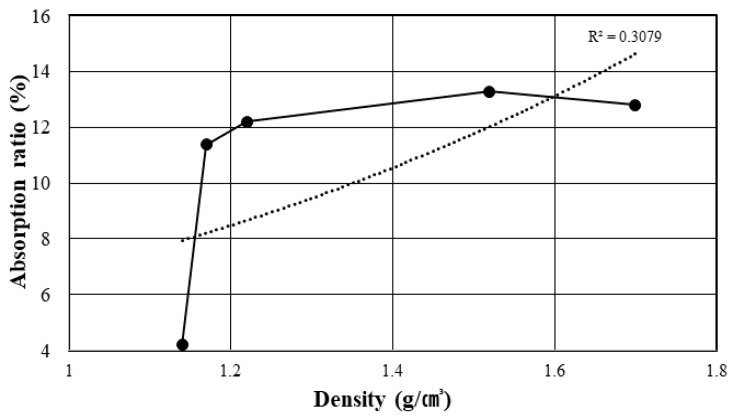
Relationship between the density and the absorption rate of dBA.

**Figure 6 materials-14-05291-f006:**
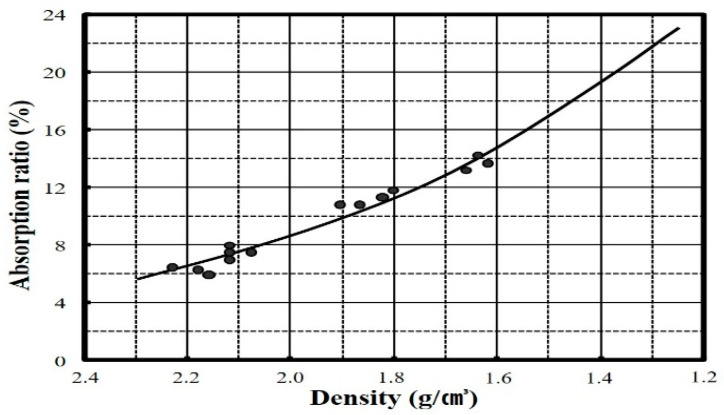
Relationship between the density and the absorption rate of a general aggregate (natural river aggregate or crushed aggregate). Source: Kim, H.S. A Study on the Quality Improvement of Recycled Fine Aggregate using Neutralization and Low Speed Wet Abraser. Notification on 2011-02; Kongju National University: Cheonan, South Korea, 2011.

**Figure 7 materials-14-05291-f007:**
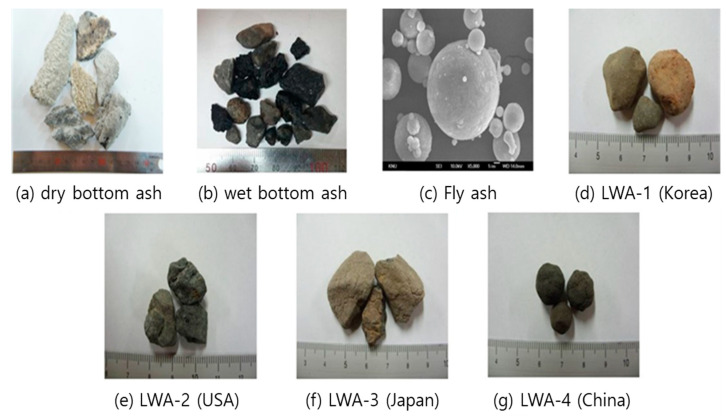
Shape of the test samples.

**Figure 8 materials-14-05291-f008:**
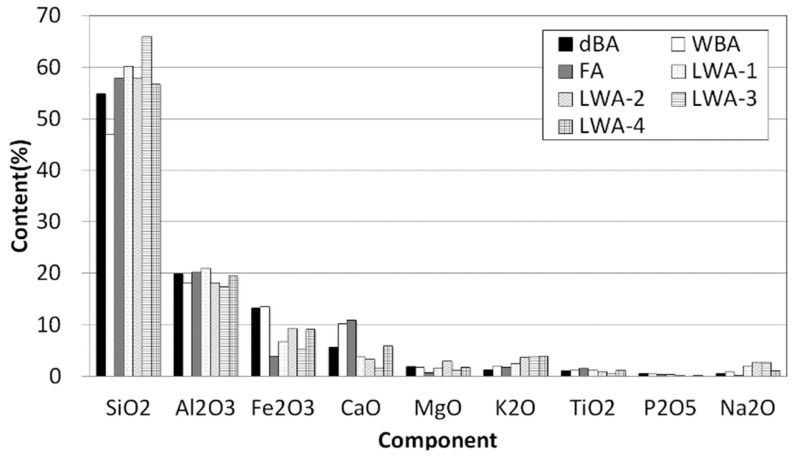
Major oxide composition of the test materials.

**Figure 9 materials-14-05291-f009:**
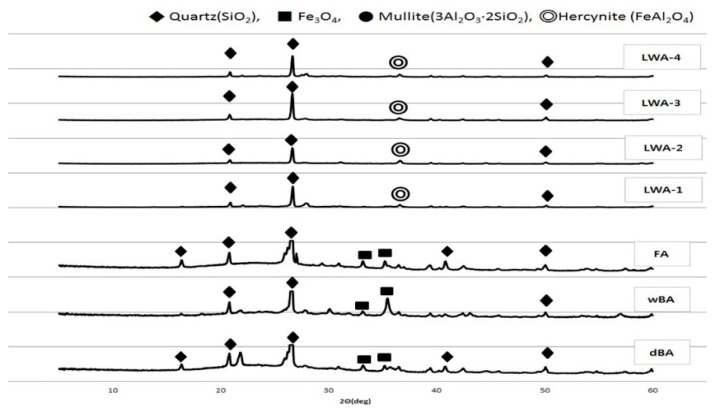
Mineral composition of the test materials.

**Figure 10 materials-14-05291-f010:**
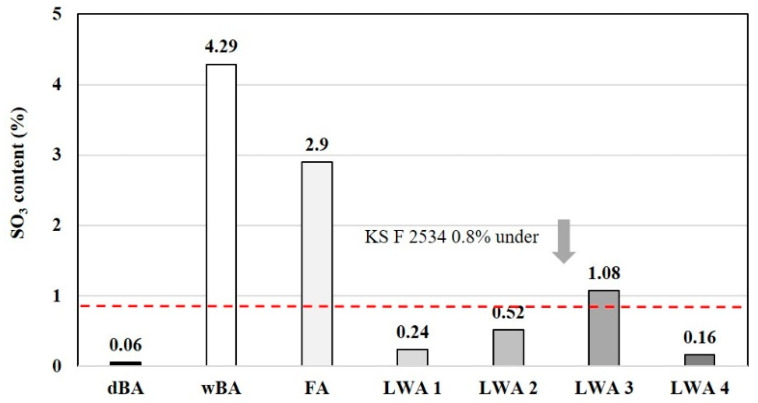
SO_3_ contents of the test materials.

**Figure 11 materials-14-05291-f011:**
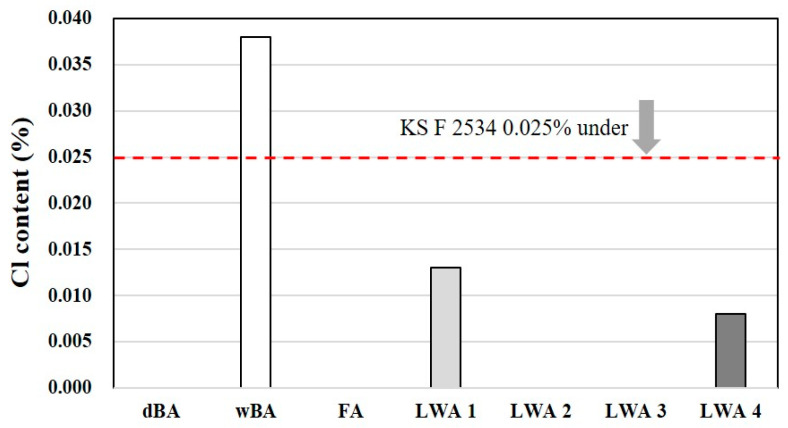
Cl contents of the test materials.

**Figure 12 materials-14-05291-f012:**
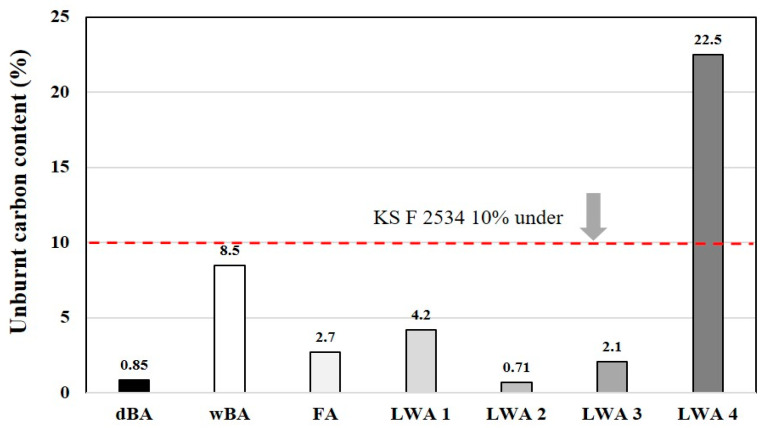
Unburnt carbon of dBA.

**Figure 13 materials-14-05291-f013:**
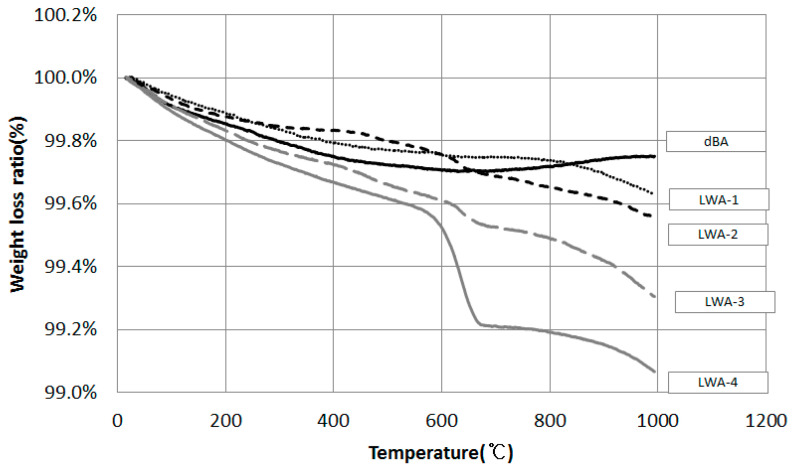
DT-TGA of coal ash and LWA.

**Figure 14 materials-14-05291-f014:**
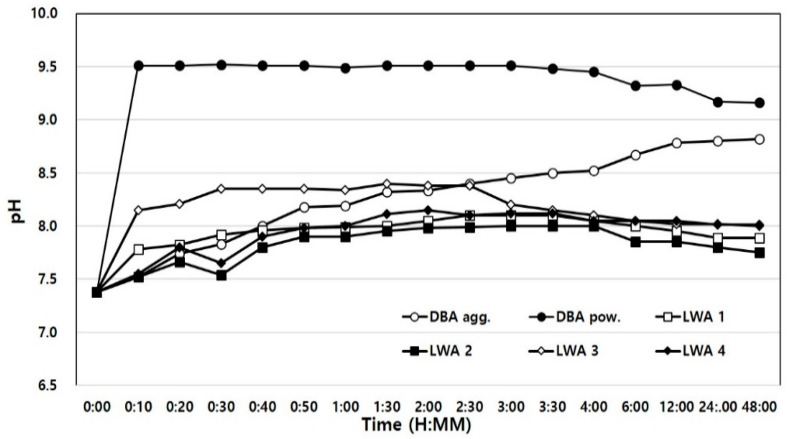
Time-dependent pH variance in tap water.

**Figure 15 materials-14-05291-f015:**
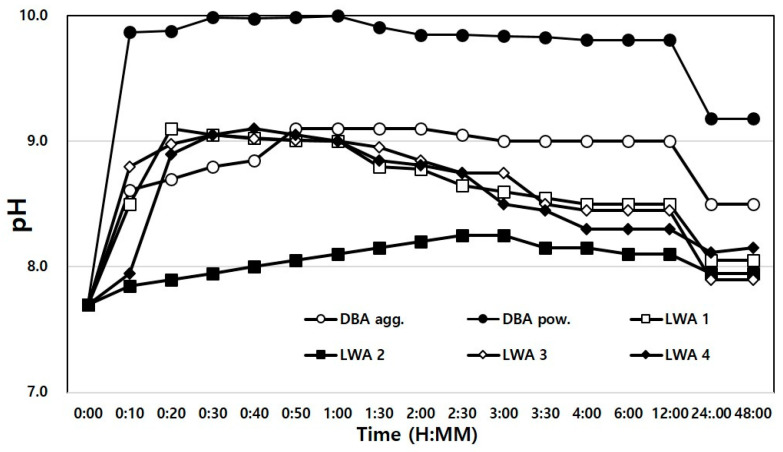
Time-dependent pH variance in distilled water.

**Table 1 materials-14-05291-t001:** Physical properties of dBA according to size.

Aggregate Size (mm)	Density (g/cm^3^)		Absorption (%)	Bulk Density (kg/m^3^)	Percentage of Absolute Volume (%)
	OD *	SSD **			
13	1.09	1.14	4.21	437.7	40.1
10	1.05	1.17	11.38	437.2	41.6
5	1.08	1.22	12.20	432.0	39.8
2.5	1.34	1.52	13.29	497.1	37.1
1.2	1.51	1.70	12.79	585.1	38.8

* OD: Oven Dry condition of dBA. ** SSD: Saturated Surface-dry condition of dBA.

**Table 2 materials-14-05291-t002:** Comparison of various aggregates.

Type of Aggregates	Advantages	Drawbacks
dBA	Low unburned carbon content Low chloride content Low SO_3_ content Open pore → can be applied to artificial ground or insulation materials	Irregular shape Structural weakness High absorption Large particles → need to crush
wBA	-	High absorption Low unburned carbon content (~25%) High chloride content → steel corrosion
Artificial light weight aggregate	Spherical shape Uniform particle size	High manufacturing cost Greenhouse gas emissions
Natural aggregate	Long-term usage experience Relatively high quality	Environmental problems Resource depletion

**Table 3 materials-14-05291-t003:** Experimental plan.

ID	Full Name of Light Weight Aggregate	Test Items
dBA	Bottom ash discharged from dry process	- Oxide composition by XRF * - Mineralogical analysis by XRD ** - Chloride content - Unburnt carbon - Potential of hydrogen - Heavy metal leaching test - Minor inorganic compounds by ICP *** - Heating loss by DT-TGA ****
wBA	Bottom ash discharged from wet process	
FA	Fly ash	
LWA-1	Artificial light aggregate from Korea	
LWA-2	Artificial light aggregate from USA	
LWA-3	Artificial light aggregate from Japan	
LWA-4	Artificial light aggregate from China	

* XRF: X-ray fluorescence, ** XRD: X-ray Diffraction. *** ICP: Inductively Coupled Plasma, **** DT-TGA: Thermogravimetry-Differential thermal analysis.

**Table 4 materials-14-05291-t004:** Experimental plan and the physical properties of various artificial aggregates.

ID	Density (g/cm^3^)		Absorption (%)	Bulk Density (kg/m^3^)	Percentage of Absolute Volume (%)
	OD	SSD			
dBA	1.72	1.76	12.11	732.8	45.52
wBA	1.67	1.73	12.34		
FA	2.10	-	-	-	-
LWA-1	1.70	1.88	10.16	1029.1	60.54
LWA-2	1.47	1.51	2.36	766.2	52.12
LWA-3	1.38	1.50	8.89	738.4	53.50
LWA-4	1.39	1.53	10.15	767.0	55.18

Source: Kim, J.M. Development for Environment Friendly Construction Material having Low Density Using the Dry Processed Bottom Ash. Notification on 2011-06; Ministry of Environment: Seoul, South Korea, 2016.

**Table 5 materials-14-05291-t005:** Trace elements of coal ash and LWA (unit: %).

	MnO	SrO	BaO	Cl	Cr_2_O_3_	ZrO_2_	NiO
dBA	0.209	0.181	0.129		0.066	0.066	0.018
wBA	0.113	0.112	0.134	0.013	0.034	0.050	0.014
FA	0.190	0.025	0.081		0.038	0.024	0.009
LWA-1	0.059	0.019			0.020	0.023	0.006
LWA-2	0.143	0.029	0.062	0.009	0.033	0.037	0.013
LWA-3	0.145	0.099	0.140	0.038	0.042	0.035	0.031
LWA-4		0.061		0.098		0.076	

**Table 6 materials-14-05291-t006:** Heavy metal content analysis type—I.

ID	Test Items (mg/L)					
	Pb	Cd	Cr6^−^	Cu	Hg	As
Standard *	3	0.3	1.5	3	0.005	1.5
dBA	N.D	N.D	N.D	N.D	N.D	N.D

* Official Wastes Test Method; Korea ministry of environment Regulation.

**Table 7 materials-14-05291-t007:** Heavy metal content analysis type—II.

ID	Test Items (mg/L)					
	Pb	Cd	Cr6^−^	Cu	Hg	As
Standard	200	4	5	150	4	25
dBA—2.5 mm size	8.04	0.27	N.D	27.7	0.01	1.61
dBA—5 mm under	6.86	N.D	N.D	6.23	N.D	N.D

**Table 8 materials-14-05291-t008:** ICP analysis.

	Alkaline Metals and Earth Metals (mg/L)				Heavy Metal (mg/L)					
	K	Mg	Ca	Na	Pb	Cd	Cr	Cu	Hg	As
dBA	1.2	34,770	10,775	22.4	0.1	ND *	1.0	515	ND	ND
wBA	0.2	5552	2223	6.3	1.3	ND	0.1	56	ND	ND
FA	0.1	9235	9552	4.9	0.0	ND	0.3	178	ND	ND
LWA-1	0.1	458	134	0.9	0.1	ND	0.6	260	ND	ND
LWA-2	4.5	673	2942	1.2	0.1	ND	0.9	174	ND	ND
LWA-3	5.1	4277	2419	1.3	0.1	ND	0.4	76	ND	ND
LWA-4	5.1	12,052	5880	28.1	0.2	ND	0.9	294	ND	ND
ST1 **	-	-	-	-	200	4	5	150	4	25
ST2 ***	-	-	-	-	400	10	15	500	10	50
ST3 ****	-	-	-	-	700	60	40	2000	20	200

* ND: Not Detected, ** ST1: Soil pollution concern Standard 1 area, *** ST2: Soil pollution concern Standard 2 area, **** ST3: Soil pollution concern Standard 3 area.

## Data Availability

The data presented in this study are available on request from the corresponding author.

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
