# Peer review of "An Evaluation of the Physical and Chemical Stability of Dry Bottom Ash as a Concrete Light Weight Aggregate"

_materials, 2021, doi:10.3390/ma14185291_

Round 1

Reviewer 1 Report

Interesting paper about the characterization of a dry process bottom ash (dBA) in order to assess its usefulness as a lightweight aggregate for construction purposes.

This submission was made following a previous one (Materials - 1213928). I can see that the major part of the suggestions issued by the reviewers were accepted by the authors.

However, I’d like to see (where appropriate) a table with all the known advantages and drawbacks of this “ash” (when compared with other artificial/natural aggregates).

As I already wrote in the previous review, this study deserves continuity with a further evaluation on this dBA applied in a sustainable concrete. You can also give note about some possible future works.

Many of the references have 10 or more years old. In recent years, several studies have been published on this topic, such as:

    “Fly Ash-Based Eco-Efficient Concretes: A Comprehensive Review of the Short-Term Properties”. By: Amran, Mugahed; Fediuk, Roman; Murali, Gunasekaran; Avudaiappan, Siva; Ozbakkaloglu, Togay; Vatin, Nikolai; Karelina, Maria; Klyuev, Sergey; Gholampour, Aliakbar. Materials (1996-1944). Aug2021, Vol. 14 Issue 15, p4264-4264. 1p. DOI: 10.3390/ma14154264;

    “Materials for Production of High and Ultra-High Performance Concrete: Review and Perspective of Possible Novel Materials”. By: Marvila, Markssuel Teixeira; de Azevedo, Afonso Rangel Garcez; de Matos, Paulo R.; Monteiro, Sergio Neves; Vieira, Carlos Maurício Fontes. Materials (1996-1944). Aug2021, Vol. 14 Issue 15, p4304-4304. 1p. DOI: 10.3390/ma14154304;

    “The Effect of Incorporating Industrials Wastewater on Durability and Long-Term Strength of Concrete”. By: Nasseralshariati, Ehsan; Mohammadzadeh, Danial; Karballaeezadeh, Nader; Mosavi, Amir; Reuter, Uwe; Saatcioglu, Murat. Materials (1996-1944). Aug2021, Vol. 14 Issue 15, p4088-4088. 1p. DOI: 10.3390/ma14154088.

Please, analyse and include other recent and relevant references.

Author Response

Authors did carefully revise the paper based on the reviewer’s comments. Detailed responses related to the reviewer’s comments are explained in the attached files “Authors-Response-#1.pdf, please check this files.

Reviewer 2 Report

General Comments:

     This paper deals with the experimental study on analysis of the physical and chemical properties of dry bottom ash produced using the dry method and examination of the possibility of using dry bottom ash as LWA for lightweight concrete. Especially it is considered that the originality is the eight test items of analysis on the physical and chemical properties of dry bottom ash compared with the other sort of light weight aggregate. Reviewer considered that the completeness of this paper is very higher. Then this paper provides valuable data for readers.

     However it is considered that there are several points that author should make modifications or correction.

1) Line 59 to 62:

Aer these sentence the situation of Korea? Or is it the whole world? Should describe it in more clarity.

2)Line 117 to 120:

The meaning of these sentence is incomprehensive. Are the solid content and the percentage of absolute volume same? Should explain the relationship solid content and aggregate particle size in detail.

3)Line 171 to 182:

There is no information of the solid content of other sort of light weight aggregate. It is considered that the solid content of DBA are much smaller compared with the light weight aggregate.

4) Line 536 to 539:

This sentence is not a conclusion. Especially the part of “but it will be necessary to improve the irregular shape into a regular spherical shape and apply consistent quality control“ is not a conclusion. Should correct this sentence.

5)Through the whole of paper:

It is considered that the objective of this paper deals with solution of the current situation of Korea. Then author should describe with the expression emphasized the situation of Korea as this paper is an international journal.

Therefore this paper is required to be improved to be published in the in the Journal of Materials.

Author Response

Authors did carefully revise the paper based on the reviewer’s comments. Detailed responses related to the reviewer’s comments are explained in the attached file “Authors-Response-#2.pdf", please check this file.

This manuscript is a resubmission of an earlier submission. The following is a list of the peer review reports and author responses from that submission.

Round 1

Reviewer 1 Report

This manuscript investigated the physical and chemical properties of dBA and other LWAs. The authors did several regular tests for dBA, wBA and other commercial LWAs, for example, XRD, XRF, ions leaching, and pH for an overall estimation. However, the discussion and explanation were limited and general, and the new information was rare. Furthermore, the writing quality of this manuscript is poor. The sentences are hard to read and understand. The innovation and contribution of this manuscript are quite limited. Therefore, this manuscript still needs a critical revision in language and discussion before submission. Some comments and questions are as follows:

  1. Line 61, this part should be included in the Introduction.
  2. Line 90, the test methods of density, absorption, unit volume mass should be added in the part of Materials and Methods.
  3. Line 105, it is well known that particle size can influence the properties of BA. However, various particle size ranges are usually selected to evaluate the physical and chemical characters. Not a specific particle size.
  4. Line 105, what is ‘solid content’? There is no information about the soluble agent of BA. Is it porosity?
  5. Line 105, what is OD and SSD density? Please add the related description.
  6. Line 108, what kinds of ‘general aggregates’ are selected in Fig.6? The unit of density is incorrect. Please check.
  7. Line 225, in Table 3, there is no information of ‘Aggregates size’, the absorption of dBA is 2.34%, however, it is 4.21% in Table 1. Can the authors explain this difference?
  8. Line 284, some peaks are not labeled in FA, WBA, and dBA.
  9. Line 286, how was SO3 content determined? XRF? Or leaching? Please specific it.
  10. Line 352, how was carbon determined? Please add test information.
  11. Line 385, the pH variation needs to be explained in detail. Why the powder dBA presents such a high pH? Some ions were leached out? Or some components were reacted in water? The pH at 0:00 should be the pH of tap water and distill water.
  12. Line 390, LWA-2 shows a pH<6 in the first minutes, can the authors explain this?

Author Response

Thank you for your review.
I attached the reply to the review. I would appreciate it if you could check it. 

Reviewer 2 Report

Interesting paper about the characterization of a dry process bottom ash (dBA) in order to assess its usefulness as a lightweight aggregate for construction purposes.

In general, the manuscript is well presented and I only advise the authors to improve/correct the following:

  1. Line 4: Author # 2 affiliation is missing;
  2. Lines 29/30: have you any information about the average amount of dBA produced in each power plant per year?
  3. Figure 2: these images were captured before and/or after crushing (first or second)?
  4. Figure 4: please change DBA by dBA;
  5. Table 1: please define OD and SSD in the text;
  6. Line 130: “as LWA” is repeated;
  7. Table 2: the acronyms “ICP” and “DT-TGA” are not defined in the text;
  8. Sections “3.3.1”, “3.3.2” and “3.3.7”: what standards were used?
  9. Line 160: “sulfur” or “sulphur”?
  10. Line 212: please change “1.76g/cm3” by “1.76 g/㎤”;
  11. Table 3: please include a space after each column title (before the units). Please identify the standards used;
  12. Line 230: please include a space before the word “accounts” in "while Al2O3accounts for 20%”;
  13. Line 255: “Quarts” or “Quartz”?
  14. Figure 10: the arrow and the text “KS F 2534 0.8 under” are not in the correct position;
  15. Line 333: please insert a space between “carbon” and “adsorbs”;
  16. Line 374: “wasslightly”?
  17. Lines 386/387: “care should be taken not to store dBA outdoors when it is powder form”, this finding is not a limitation?
  18. Table 5: please, include an extra line with the content requirements, for example, under the Korean standard (or other);
  19. Did you analyse the compliance of the concrete light weight aggregate, used in this study (dBA), with the COUNCIL DIRECTIVE 2013/59/EURATOM of 5 December 2013 - Official Journal of the European Union (establish reference levels for indoor radon concentrations and for indoor gamma radiation emitted from building materials) or with other similar standards?
  20. You can organize, for example, a table with the known advantages and drawbacks of this “ash”.
  21. This study deserves continuity with a further evaluation on this dBA applied in a sustainable concrete. You can also give note about some possible future works;
  22. About 22 references (in a total of 38) have 10 or more years (up to or equal to 2011). Only 7 references have less than 5 years. Can you include other recent references?

Author Response

(The authors gave the same response as above.)

Reviewer 3 Report

The article presents a very interesting topic of using waste materials as lightweight aggregate for concrete. However I do not recommend this paper for publication. Due to the large number of comments and objections, the article is not suitable for publication in its present form.

The description provided in section 2.3 does not correspond to the description in Table 1, similarly the description in section 3.4 does not correspond to the description of Table 3. 

The conclusions in rows 97 ÷ 101 do not match the data in Table 1.

The bulk density value should be given in SI units (Table 1 and 3).

The description of the SSD density should be included in the description of Figure 5.

The graph shown in Figure 6 does not match the data in Table 1.

In the descriptions, the designation WBA and DBA is used instead of wBA and dBA(Figure 4, 10, 11).

The description of the test method does not correspond to the descriptive test results, the results of the mineralogical analysis also include results for chloride content and unburned carbon. 

Incorrect section numbering, Section 4.2.5 is followed by Section 4.4.6.

The density value for wBA in Table 3 is close to the specific density value for clay ( see line 249, 250 for comparison) - in my opinion an unrealistic value.

Some values are missing in Table 3 for wBA and FA.

In section 4.2.3, the description in lines 306÷312 corresponds to wBA. 

The graph in Figure 12 is not very clear, line graphs for Facilities 7 and 8 separately would be clearer.

Figure 13 is missing data for wBA and FA.

Tables 5, 6, and 7 are not properly cited in the narrative; Tables 6 and 7 show the results of the heavy metal analysis and should be cited in Section 4.4.6. Table 6 on line 422, Table 7 on line 427. On line 435 there should be a reference to Table 5 (ICP analysis) and not to Tables 6 and 7. Of course, the numbering of the tables should be changed. 

Heavy metal leaching studies are lacking for wBA, FA, and all LWA. The justification for the lack of these studies for LWA presented in lines 415÷417 is inadequate.

The final conclusions contain only a direct reference to the results of part of the study. There is lack of general conclusions concerning the title of the article, lack of presentation of the possibility of application of dBA aggregates in the construction industry. It would be advised to present in which areas of construction it is possible to apply dBA as opposed to wBA. Additionally, a comparison of the applications of dBA and LWA should be made. 

Author Response

(The authors gave the same response as above.)

Reviewer 4 Report

In general, the paper is well written, however, after reviewing the papers, the reviewer has some comments as follows:

  1. Introduction: The authors should expand and review the application of bottom ash in concrete and mortar. There is a huge study on its application, thus please expand this application.
  2. 4: The reviewer thought that the vertical axis should be “accumulated porosity” instead of “porosity”, please consider it.
  3. Table 1: The author should make an explanation (note) for the abbreviation of OD and SSD, in addition, please use (kg/m3 or kg/dm3) instead of (kg/l).
  4. 6. If this figure was taken from a previous study, please kindly cite it.
  5. Line: 117: what is “true density”, this word seems to be not a technical term.
  6. Line 130: repeat LWA, it is not needed.
  7. Table 3: if the value of coal ash and LWA are taken from previous studies, please make the citation to indicate clearly the sources.
  8. 10: Please add discussion why WBA achieved the highest SO3 content and the content of SO3 in dBA is the lowest.
  9. Sentence 332-334: “If the loss on ignition is high, the unburned carbon adsorbs the entraining air introduced to improve the durability of the concrete, and this adversely affects the durability of the concrete.” Please kindly check this sentence.
  10. check some minor errors in English.

Regards,

Author Response

(The authors gave the same response as above.)

Reviewer 5 Report

General Comments:

I considered that the completeness of this paper is very higher. Then this paper provides valuable data for readers. However it is considered that there are several points that author should make modifications or correction.

1) Revision of the word defined by Author

It is considered that the word of “dBA” and “wBA” are not good and are strange in comparison with “FA” and “LWA” in this paper. Should change to “DBA” and “WBA”.

2) Name of instrument and measurement for analysis

Author had better not describe the company names of the instrument and measurement machine used in this text. Should write a general equipment name in this text. Author should introduce the company names to references.

3) Explanatory notes of figure 4 at line 87 

A meaning of “Open“ and “Closed” are not incomprehensive. Should describe the implication of these explanatory notes in this paper shortly.

4) Line 95 to 96

What is “general aggregate”?  It this aggregate is natural river aggregate or crushed aggregate or LWA ?  Author should describe the date of Figure 6 in detail.

5) Table 3 at line 225

“Aggregate size(㎜)” is a mistake of“ID”. Please revise it.

6) Line 296 to 297

The implication of the sentence of “This is because some of the coal ash discharged from bituminous coal-fired power plants contains sulfur trioxide.” is not incomprehensive. Please explain the meaning of this sentence in detail.

7) Figure 11 at line 303

The position of “KS F 2534 0.8”under “is strange. Please revise it.

8) Line 375 to 378

The word of “regular service water“and “service water” are revised as the word of “tap water”.

9) “5. Conclusion” form Line 453 to 477

The volume of text is redundant. Please shorten it more.

Therefore this paper is required to be improved to be published in the in the Journal of Materials.

Author Response

(The authors gave the same response as above.)

Round 2

Reviewer 3 Report

Not all previous comments have been addressed. There are still many errors in the description.   The description of the tables on line 489 is incorrect; the density units have not been corrected throughout (g/cm3, kg/m3).  Still the final conclusions contain only a direct reference to the results of part of the study. There is lack of general conclusions concerning the title of the article, lack of presentation of the possibility of application of dBA aggregates in the construction industry. It would be advised to present in which areas of construction it is possible to apply dBA as opposed to wBA. Additionally, a comparison of the applications of dBA and LWA should be made. 

Reviewer 4 Report

Thank you for the efforts from the authors, however, many revised parts, as well as the response of the authors to the reviewer, are not enough. The authors have not carefully considered the suggestions and comments of the reviewer.

For example, the authors did not expand the literature suggests by the reviewer. There are many studies using bottom ash and its application as an aggregate in the concrete field.

The authors took many data and results from the previous studies, the review has suggested making citations but the author may not cite exactly way (by quoting in explanation, this is not a way of citation ...).

Some statements were suggested by the reviewer to revise carefully but it seems that the authors did not understand or may not consider exactly. 

Thus, the reviewer would like to reject this paper due to the carelessness of the authors.

Thank you.